# Pretreatment with Betablockers, a Potential Predictor of Adverse Cardiovascular Events in Takotsubo Syndrome

**DOI:** 10.3390/biomedicines10020464

**Published:** 2022-02-16

**Authors:** Albert Topf, Moritz Mirna, Christiane Dienhart, Peter Jirak, Nina Bacher, Elke Boxhammer, Sarah X. Gharibeh, Lukas J. Motloch, Uta C. Hoppe, Michael Lichtenauer

**Affiliations:** 1Clinic for Internal Medicine II, University Hospital Salzburg, Paracelsus University Salzburg, Müllner Hauptstraße 48, A-5020 Salzburg, Austria; m.mirna@salk.at (M.M.); p.jirak@salk.at (P.J.); n.bacher@salk.at (N.B.); elke-bo@freenet.de (E.B.); sa.gharibeh@hotmail.com (S.X.G.); l.motloch@salk.at (L.J.M.); u.hoppe@salk.at (U.C.H.); m.lichtenauer@salk.at (M.L.); 2Clinic for Internal Medicine I, University Hospital Salzburg, Paracelsus University Salzburg, Müllner Hauptstraße 48, A-5020 Salzburg, Austria; c.dienhart@salk.at

**Keywords:** Takotsubo syndrome, betablocker, adverse cardiovascular events

## Abstract

**Introduction**: Treatment with betablockers is controversial in Takotsubo syndrome (TTS); however, many physicians intuitively initiate or continue betablocker therapy in these patients. The effect of preadmission betablocker use on adverse cardiovascular events has not been studied in the literature. **Methods:** To investigate this issue, we evaluated clinical complications, defined by the endpoint of occurrence of hemodynamically relevant arrythmia, cardiac decompensation, and all-cause adverse cardiac events, during hospitalization, in 56 patients hospitalized for TTS between April 2017 and July 2021. We compared the risk of adverse cardiovascular events between patients with preadmission betablocker therapy and those without preadmission betablocker therapy. Pretreatment betablocker therapy was defined as daily betablocker intake for more than a week including day of admission. **Results****:** TTS patients taking preadmission betablockers had a significantly increased risk of all-cause complications relative to patients without betablockers in preadmission medication ((52.0% vs. 19.4%, *p* = 0.010; OR 4.5 (95% Cl 1.38–14.80)). Furthermore, TTS patients already taking betablockers on admission showed a statistically significant increased risk of cardiac decompensation when compared to patients without pretreatment with betablockers (*p* = 0.013). There were no significant differences in patient characteristics in patients who were taking beta blockers as an adjunct therapy prior to admission for TTS relative to those who were not. There is however an increase in comorbidities, hypertension, and atrial fibrillation, in past medical history in patients taking a preadmission betablocker. The difference is related to therapeutic applications for beta blockers and was not significant based on endpoints of our study. **Conclusions:** Preadmission betablocker treatment was associated with a 4.5 times higher risk of adverse cardiac events. This increased risk of all-cause complications and of cardiac decompensation within the acute phase of TTS is presumably due to the negative inotropic effects of betablockers and upregulation of β-adrenergic receptors in patients with chronic betablocker therapy.

## 1. Introduction

While Takotsubo syndrome (TTS) is considered rare, it has an incidence of up to 7.5% in the female population with a preponderance for postmenopausal women. It presents with symptoms similar to an acute myocardial infarction, but is considered a heart failure condition. Three percent of all suspected acute coronary syndromes (ACS) are caused by TTS [1,2].

TTS causes reversible wall motion abnormalities involving apical, mid-ventricular, or basal segments of the left ventricle and is defined by acute left ventricular dysfunction in the absence of a significant coronary stenosis. [1,3]. While 96% of cases show a nearly full recovery with resolution of the wall motion abnormalities within a few days, TTS can be potentially life threatening with a mortality rate of one to two percent in the acute phase [4,5]. One in five patients develops congestive heart failure that requires diuretic drug use with intravenous therapy. Of these, some also require additional ventilation support. Life-threatening ventricular arrhythmias are observed in 8.6% of TTS patients. Left ventricular wall rupture, cardiogenic shock, and thrombosis are also reported [6].

The pathogenic mechanism behind TTS is not well defined, but is thought to be due to an inadequate response of the heart muscle to excessive epinephrine release [4]. It is often associated with preceding emotional and physical stress factors.

Betablockers are used as an evidence-based treatment for heart failure with reduced ejection fraction (HFrEF), ACS, anginal symptoms, and prevention of sudden cardiac death [7,8,9,10]. However, based on the analysis of retrospective studies, the use of betablockers in the short- and long-term management of TTS patients remains controversial [11]. There is some evidence for beneficial effects of short acting betablockers in the presence of hemodynamic relevant left ventricular outflow tract obstruction (LVOT) [12]. However, the effect of pretreatment with betablockers on the clinical outcome of TTS has not been investigated previously. 

The purpose of this study was to analyze the effect of preadmission betablocker use on the presentation and clinical course of TTS.

## 2. Materials and Methods

### 2.1. Patients and Controls

In this prospective study, we included 56 patients with TTS, who were in treatment at the University Hospital of Salzburg between April 2017 and July 2021. Fulfillment of the Mayo Clinic Diagnostic Criteria for TTS was necessary for enrollment. According to the Mayo Clinic Diagnostic Criteria, TTS is defined as a transient ventricular dysfunction with new ECG abnormalities or troponin elevation in the absence of obstructive coronary artery disease, pheochromocytoma, and myocarditis [13]. We compared the risk of adverse cardiovascular events, defined by three endpoints, between patients with preadmission betablocker therapy and those without preadmission betablocker therapy. Pretreatment betablocker therapy was defined as patients who had been on a daily betablocker intake for more than a week including the day of admission. Patients on pretreatment with betablockers remained on therapy until discharge. We recorded data on clinical presentation, precipitating factors, cardiovascular risk factors, medications, and demographics at the time of admission to hospital. Blood analysis was performed at the time of admission to hospital as well. Adverse cardiac events were also documented during admission with a focus on the three predefined study endpoints. Patients were monitored with 24 h of ECG monitoring while in intensive care. The majority of patients also had a 24 h ECG during follow-up post hospitalization. Supra- and ventricular arrythmias, which caused hemodynamic compromise, were considered as an endpoint. Documented arrhythmias with defined hemodynamic comprise according to the ERC guidelines, which necessitated immediate pharmacological therapy or electrical cardioversion, were recorded during admission. Furthermore, patients with hemodynamic relevant supraventricular tachycardia (SVT) were only enrolled in this study if similar episodes of SVT had not been previously documented; this was undertaken to ensure that only patients with Takotsubo induced supraventricular arrythmia would be enrolled.

Cardiac decompensation, which was defined as clinical and/or radiographic signs of fluid retention that required diuretic drug use and/or ventilation therapy, was considered the second endpoint. X-rays were analyzed by an independent and experienced radiologist. Decision for diuretic drug use and/or ventilation therapy was made by two experienced cardiologists.

The third endpoint encompassed all-cause complications including cardiac decompensation and/or arrythmias and/or cardiogenic shock and/or left ventricular wall rupture or thrombosis and/or cardiovascular death within the acute phase of TTS. All-cause complications and adverse cardiac events are used synonymously throughout the manuscript. 

The study was performed in accordance with the Declaration of Helsinki and Good Clinical Practice (local ethics committee approval: 415-E/2230/10-2018, approval April 2017). Written informed consent was given by all patients prior to enrollment. 

### 2.2. Statistical Analysis

A Kolmogorov–Smirnov test was performed to assess the distribution of data in the study population. As most parameters and biomarker concentrations were not normally distributed, all are depicted as median and interquartile range (IQR). Median values between groups were compared by Mann–Whitney-U test. Nominal parameters were compared by Chi Square test. Binary logistic regression was used to assess the influence of patients’ characteristics and premedication with the risk of adverse cardiac events. To prevent overfitting, only three statistically significant covariates were included in the multivariate binary logistic regression model. Elimination criterion was a *p*-value above 0.10. A *p*-value < 0.05 was considered to be statistically significant.

SPSS (22.0, SPSS Inc., Chicago, IL, USA) was used to perform the statistical analysis.

## 3. Results

### 3.1. Baseline Characteristics

Baseline characteristics of TTS patients are shown in Table 1. TTS patients had a median age of 70.0 years. There was no known history of significant coronary artery disease or heart failure in our patients. A total of 21 out of 56 patients had no coronary artery disease in coronary angiography; the remainder had non-significant coronary artery stenosis; 11 out of 56 patients had a preceding emotional trigger. With respect to comorbidities, smoking was prevalent in 16 out of 56 TTS patients.

As expected, TTS patients showed a reduced left ventricular ejection fraction on admission (median: 40.0%, IQR: 35.0–45.5%). The median systolic blood pressure (SBP) was 130.5 and the median diastolic blood pressure (DBP) 83.0 mmHg.

Pro-BNP levels were elevated (median 2747.0 pg/mL, IQR: 557.0–5396.0 pg/mL). Patients also showed increased high sensitivity troponin levels (median 192.0 pg/mL, IQR: 57.0–455.0 pg/mL. 

Cardiac decompensation with either radiographic and/or clinical signs of fluid retention was found in 20 out of 56 patients. Ten patients had arrhythmias with hemodynamic compromise. Four patients had sustained ventricular tachycardia, of which two had Torsade de Pointes. One patient needed permanent pacemaker implantation due to third degree AV block. Five patients developed de novo supraventricular arrythmias (1× atrial flutter, 2× atrial fibrillation), which required urgent cardioversion due to hemodynamic deterioration during rhythm monitoring on the intensive ward. Two patients with sudden onset of hemodynamic compromise due to atrioventricular nodal reentrant tachycardia were converted using adenosine. Considering combined all-cause complications, adverse cardiac events were observed in 25 out of 56 TTS patients. There were no fatalities among the patients included in this study. 

When considering preadmission medication, 19 out of 56 patients were on prior betablocker therapy, most frequently as an adjunct hypertensive drug therapy but also for rate control in patients with atrial fibrillation. Bisoprolol (68.4%) was the most frequently used beta blocker with a median dosage of 2.5 mg (IQR 2.5–5 mg), followed by nebivolol (21.1%) with a median dosage of 5 mg (IQR 3.1–5 mg), and carvedilol (10.5%) with a median dosage of 37.5 mg. 

### 3.2. Comparison of Patient Characteristics between Patients with Prior Betablocker Use Versus no Prior Betablocker Use

There was no statistically significant difference in median age of TTS patients with and without prior betablocker use. (Table 2). Patients’ comorbidities, including smoking status, BMI, presence of oncologic disease, and dyslipidemia, did not differ among the two groups. Comorbidities of hypertension and a history of atrial fibrillation were more frequent in TTS patients with preadmission betablocker therapy. There were no statistically significant differences in measured parameters of myocardial damage/dysfunction and LVEF between the two groups. While both median values were still within the normal range, creatinine levels and HbA1c were statistically significantly higher in patients with previous betablocker use. 

### 3.3. Pretreatment with Betablockers as Potential Predictor of Adverse Cardiac Events within the Acute Phase of TTS

There was a significant difference in all-cause complications in patients pretreated with betablockers and those who were not (*p* = 0.010; OR 4.5 (95% Cl 1.38–14.80), see Table 3 and Table 4). Pretreatment with ACE inhibitors/AT-1 inhibitors (*p* = 0.074), anticoagulation (*p* = 0.272), SSRI/SNRI antidepressants (*p* = 0.446), diuretics (*p* = 0.217), and inhalation therapy (*p* = 0.346) were not associated with increased risk of all-cause complications.

When considering patients’ characteristics, BMI (*p* = 0.972), smoking status (*p* = 0.610), SBP (*p* = 0.760), DBP (*p* = 0.625), LDL levels (*p* = 0.236), and HbA1c levels (*p* = 0.932) did not differ among TTS patients with and without complications. History of atrial fibrillation (AF) (*p* = 0.251) and of stroke (*p* = 0.205), as well as female sex (*p* = 0.823), were not predictive for adverse events. Based on median age, TTS patients with complications were significantly older than TTS patients without complications (*p* = 0.045).

Left ventricular ejection fraction was significantly lower in TTS patients with all-cause complications compared to patients without complications within the acute phase (*p* = 0.019), but there was no significant difference in left atrial size (*p* = 0.981) and in E/E’ (*p* = 0.525). The length of QTc interval (*p* = 0.554) and heart rate (*p* = 0.112) also did not show a statistically significant difference.

Whereas pro-BNP levels differed significantly between TTS patients with and without all-cause complications (*p* = 0.026), hs-troponin (*p* = 0.179), creatinine (*p* = 0.888), and CRP levels (*p* = 0.402) did not show a difference.

Using univariate regression analysis, prior betablocker therapy was predictive for all-cause complications (see Table 4). This association remained statistically significant in a multivariate regression model after correction for age and systolic left-ventricular ejection fraction.

### 3.4. Preadmission Betablocker Therapy as Potential Predictor of Cardiac Decompensation within the Acute Phase of TTS

TTS patients with prior betablocker therapy showed a significantly increased risk of cardiac decompensation when compared to patients without pretreatment with betablockers (*p* = 0.013, see Table 5). There was no difference in the risk of cardiac decompensation in patients with and without pretreatment with ACE/AT-1 inhibitors (*p* = 0.053), anticoagulation (*p* = 0.362), SSRI/SNRI antidepressants (*p* = 0.119), diuretics (*p* = 0.069), and inhalation therapy (*p* = 0.342).

When considering patients’ characteristics, there was no association of TTS patients’ BMI (*p* = 0.184), smoking status (*p* = 0.860), SBP (*p* = 0.661)/DBP (*p* = 0.697), LDL levels (*p* = 0.285), and HbA1c levels (*p* = 0.526) with the risk of cardiac decompensation within the acute phase of TTS. CRP (*p* = 0.215) and creatinine levels (*p* = 0.077) did not statistically differ when considering risk of cardiac decompensation. A history of stroke (*p* = 0.333) and florid oncologic disease (*p* = 0.206), as well as female sex (*p* = 0.699) did not show a statistically significant difference among the investigated characteristics. 

Neither the QTc interval (*p* = 0.426), nor heart rate (*p* = 0.300) was significant with regard to the risk of cardiac decompensation.

Using univariate regression analysis, prior betablocker therapy was predictive for cardiac decompensation (see Appendix A). This association did not remain statistically significant in a multivariate regression model after correction for age and systolic left-ventricular ejection fraction (*p* = 0.073).

### 3.5. Preadmission Betablocker Use as Potential Predictor of Arrythmia within the Acute Phase of TTS

TTS patients with pretreatment with betablockers did not show a significant difference in risk of arrhythmia when compared to patients without pretreatment with betablockers (26.3% vs. 13.5%; *p* = 0.236, see Table 6). There was no significant difference in the risk of arrhythmia in patients with and without pretreatment with ACE/AT-1 inhibitors, anticoagulation, SSRI/SNRI antidepressants, diuretics, and inhalation therapy.

There was no association between patients’ characteristics, including BMI, smoking status, systolic/diastolic blood pressure, LDL and HbA1c levels, and the risk hemodynamically relevant arrythmias within the acute phase of TTS.

Routine transthoracic echocardiography parameters, such as left atrial size, E/E’, and left ventricular ejection fraction, were not predictive for the risk of arrhythmias. High sensitivity troponin, BNP, CRP, and creatinine levels did not affect the occurrence of rhythm events. 

Using univariate regression analysis, prior betablocker therapy was not predictive for arrythmia (see Appendix A). This association did not remain statistically significant in a multivariate regression model after correction for age and systolic left-ventricular ejection fraction.

## 4. Discussion

Despite TTS often-perceived benign nature, with functional recovery within a few weeks in almost 96% of cases, life threatening complications and even fatalities can be expected in the early course of the disease [14]. Therefore, observation on an intensive care unit is common clinical practice in the acute setting of TTS [15]. Cardiac decompensation, hemodynamically relevant arrhythmias, and cardiogenic shock may make a transfer to an intensive care ward necessary for a ventilation therapy or even extracorporeal circulation therapy.

Concomitant infection or neurological events, recent surgical procedures, physical activities, and severe hypoxia are factors that have been reported to worsen the short-term outcome for TTS patients. Age > 70 years, diabetes mellitus, left ventricular ejection fraction < 30%, and shock at admission have been reported to affect long-term prognosis [16]. In keeping with previous research, patients’ age, left ventricular ejection fraction, and subsequent BNP levels were indicative for the prediction of all-cause complications in our study. However, the effect of preadmission medication on the short-term outcome has not been thoroughly investigated to date. 

In the International Expert Consensus Document on Takotsubo Syndrome, the role of initiation of betablocker therapy on the short-term and long-term outcomes of TTS patients has been commented on, but the role of preadmission betablocker therapy has so far not been the focus of prior studies [17]. There has been only one study, Palla et al., investigating the role of preadmission betablocker treatment on the levels of cardiac enzymes, left ventricular end diastolic pressure, and on left ventricular ejection fraction in TTS patients at the time of hospital admission [18].

The use of betablockers in TTS is a controversial issue. As catecholamine levels are elevated in TTS, betablocker therapy until LVEF normalization seems reasonable. However, strong evidence to support this practice is lacking. Data showing the benefit of betablocker use in the acute phase of TTS are limited to an animal study showing that apical ballooning is attenuated after the administration of metoprolol [19], and to some case reports or observational studies [20,21,22]. Others found that betablockers could carry a risk of adverse effects in TTS patients. A multicenter study, involving 36 patients, showed that 30-day treatment with an oral betablocker initiated at the time of diagnosis, provided no advantage in improvements of left ventricular ejection fraction [23]. Kummer et al. reported, that betablocker therapy did not ameliorate long-term survival in TTS patients [24]. Similarly, in an analysis of a TTS registry, Templin et al. showed comparable death rates at 1 year whether patients with TTS are treated with betablockers or not [25]. An observational study by Isogai et al. revealed that 30-day mortality, as a short-term outcome surrogate parameter, was unaffected by betablocker therapy [26]. Three meta-analyses by Singh et al., Bonacchi et al., and Santoro et al. concluded that betablockers do not prevent TTS recurrence [27,28,29]. Better evidence for short acting betablocker therapy is given in TTS patients with cardiogenic shock and the presence of hemodynamic relevant left ventricular outflow tract obstruction (LVOTO). Nevertheless, while they may improve LVOTO, betablocker therapy is generally contraindicated in acute and severe heart failure with low LVEF, hypotension, and in patients with bradycardia [15]. There is also expert opinion that, as TTS is considered to be primarily self-limiting, unnecessary treatment should be avoided [30]. In the ESC International Expert Consensus Document on Takotsubo Syndrome, there is the recommendation that the management of TTS is primarily empirical and needs to be individualized for each patient due to the lack of prospective randomized clinical trials. 

Given the controversy surrounding the use of betablockers in the management of TTS, our 56 TTS patient study identifies prior betablocker use as a possible additional risk factor for adverse cardiac events within the acute phase of TTS. Furthermore, our study may provide novel insights into the efficacy of betablocker treatment, as delayed initiation of betablockers was suspected by some authors as a limitation of its effect [26]. 

Fifty-six TTS patients were prospectively enrolled to investigate the effect of a preadmission betablocker treatment on three clinical endpoints. When considering the most important clinical characteristics for adverse cardiac events within the acute phase of TTS, there was no significant difference among patients with preadmission betablocker use and those without, except higher occurrence of hypertension and history of atrial fibrillation in those patients already using betablockers. Adjunct blood pressure therapy was the most relevant indication for betablocker intake. However, hypertension was not associated with increased risk of adverse cardiac events in TTS. Although creatinine and HbA1c levels were statistically significantly different between groups, the median was still well within the normal range. Therefore, preadmission betablocker therapy may be considered a relatively unbiased potential predictor of prospective adverse events within the acute phase of TTS. In our study, pretreatment with a betablocker was associated with a 4.5 times higher risk of all-cause complications and a similarly high risk of cardiac decompensation requiring diuretic and/or ventilation therapy. Therefore, pretreatment with betablockers in TTS patients may necessitate prolonged monitoring at an intensive care ward and a longer hospitalization. Consequently, given the detrimental data on the use of betablockers in short- and long-term management, as well as in the prevention of recurrent TTS, an even more restrictive use of betablockers might be reasonable after the analysis of our study results. Moreover, as 96% of the cases show a full recovery with resolution of wall motion abnormalities within a few days, the main purpose of our management should focus on the prevention of adverse events. Therefore, a better identification of TTS patients at risk of adverse events is necessary. In consideration of our study results, TTS patients taking betablockers prior to admission should be perceived to be at a 4.5 times higher risk of cardiac adverse events. 

The pathophysiological mechanism behind the detrimental effect of preadmission betablocker use and risk of adverse cardiac events in TTS patients may be explained by the effect of betablockers on beta-receptor channels within the endogenous catecholamine surge in TTS. The main hypothesis with respect to the pathophysiology of TTS is that patients suffer from catecholamine-induced cardiotoxicity. Plasma catecholamines above supra-physiological levels are hypothesized to cause an overactivation of the β1-adrenoceptor-protein kinase-A pathway, which causes cardiomyocyte necrosis via intracellular calcium overload and oxidative stress. Catecholamines, secernated directly into the myocardium via sympathetic nerves, are suspected to be more poisonous than catecholamines, released via the bloodstream [31]. High epinephrine levels, but not norepinephrine levels, may result in a switch from the protein coupled to β-adrenoceptors with a stimulatory protein (Gαs) to an inhibitory protein (Gαi) [32]. A 35-fold higher affinity for epinephrine than for norepinephrine is reported for β2-adrenoceptors. The typical apical ballooning type of TTS may therefore be the result of lower density of sympathetic nerve terminals (which release norepinephrine) in the apical LV segments and a higher concentration of β2-adrenoceptors in the apical area [33]. From experimental studies, it was reported that treatment with specific β2-adrenoreceptor blockers did not appear to be a therapeutically successful, because of an exacerbation of the catecholamine-induced negative inotropic effect depending on the level of β2-adrenoceptors-Gi agonism. For example, due to its higher β2-adrenoceptors-Gi agonism, propranolol significantly enhanced and prolonged the negative inotropic effects of epinephrine at both cardiac apex and base. In comparison, carvedilol had little effect on the apex but converted the base from a positive to a significant negative contractile response, since it is a weaker β2-adrenoceptors-Gi agonist. In contrast, the β1-adrenoceptors-selective blocker, bisoprolol, reduced the positive effect of epinephrine at the base, but there was no effect on the apical catecholamine-induced effect [32]. These data from experimental methods support the results from our study, whereby preadmission betablocker use is associated with more adverse cardiac events within the acute phase of TTS. Presumably, betablocker therapy may be associated with a higher risk of all-cause complications and cardiac decompensation within the acute phase of TTS via hemodynamic impairment due to negative inotropic effects of betablockers. Furthermore, patients on chronic betablocker therapy develop increased sensitivity and upregulation of β-adrenergic receptors, and therefore the effect of a sudden catecholamine surge in TTS patients could be exaggerated in these patients [34]. When considering treatment of sympathetic overdrive, α-lipoic acid demonstrated an improvement of sympathetic heart function in TTS patients. [35] Similarly, TTS patients with hyperglycemia were reported to present with higher sympathetic overactivity and therefore with worse prognosis. [36]; [37]. Therapies, targeting sympathetic overdrive might reveal novel therapeutic options in TTS patients.

## 5. Conclusions

Based on our results, we conclude that preadmission betablocker therapy increased the risk of adverse cardiac events 4.5-fold. The risk of cardiac decompensation was also similarly elevated in our study population. The increased risk of all-cause complications and cardiac decompensation within the acute phase of TTS is presumably due to the negative inotropic effects of betablockers and the upregulation of β-adrenergic receptors in patients with chronic betablocker therapy.

## 6. Limitations

Among the limitations of the present study are the small cohort. Another bias is that there is unequal distribution between TTS patients with preadmission betablocker intake and those without. Patients in our cohort frequently took beta blockers as an adjunct hypertensive therapy. Thus, there was an increased number of patients with hypertension as a comorbidity in the preadmission betablocker group. Large-scale studies are required to confirm the results of the present study.

## Figures and Tables

**Table 1 biomedicines-10-00464-t001:** Baseline characteristics of all TTS patients, given as median and IQR or n (%).

	TTS	
	**Median or n (%)**	**IQR**
Age (years)	70.0	62.0–78.0
Sex (female)	92.8	
BMI (kg/m^2^)	25.2	21.8–29.3
Smoking	28.6	
Oncologic disease	12.5	
Stroke	7.1	
History of AF	10.7	
EF (%)	40.0	35.0–45.5
SBP (mmHg)	130.5	117.8–156.0
DBP (mmHg)	83.0	70.0–92.8
Heart rate (bpm)	78.0	68.0–90.0
QTc (ms)	457.0	436.5–490.5
Creatinine (mg/dL)	0.8	0.7–0.9
LDL (mg/dL)	92.0	75.5–123.5
CRP (mg/L)	0.5	0.1–1.1
HbA1c (%)	5.4	5.2–5.8
(hs) Troponin (pg/mL)	192.0	57.0–455.0
Pro-BNP (pg/mL)	2747.0	557.0–5396.0
Arrythmia	17.9	
Cardiac decompensation	35.7	
All-cause complications	44.6	
Betablocker	33.9	
ACE/AT-1 inhibitor	42.9	
Diuretics	17.9	
Inhalation therapy	23.2	
Anticoagulation	14.3	
SSRI/SNRI	23.2	

AF (Atrial Fibrillation), BMI (Body Mass Index), CRP (C Reactive Protein), DBP (Diastolic Blood Pressure), EF (Ejection fraction), HbA1c (hemoglobin A1c), SBP (Systolic Blood Pressure), LDL (low density lipoprotein), SSRI/SNRI (Selective Serotonin Reuptake Inhibitor/selective noradrenaline reuptake inhibitor).

**Table 2 biomedicines-10-00464-t002:** Comparison of parameters of clinical presentation, precipitating factors, cardiovascular risk factors and demographics between patients with and without preadmission betablocker use by Mann–Whitney-U test or, if nominal, by Chi Square test.

	TTS Patients with Betablocker Therapy (*n* = 19)		TTS Patients without Betablocker Therapy (*n* = 37)		*p* =
	Median or n (%)	IQR	Median or n (%)	IQR	
Age (years)	76.0	64.0–78.0	67.0	61.0–78.0	0.068
Sex (female)	89.5		94.6		0.481
BMI (kg/m^2^)	26.7	21.7–29.3	23.9	21.8–29.7	0.528
Smoking	21.1		32.4		0.372
Oncologic disease	21.1		8.1		0.166
History of AF	26.3		2.7		0.007
Hypertension	84. 2		59.5		0.009
History of stroke	21.1		0.0		0.004
Non-significant Coronary artery disease	42.1		35.1		0.910
EF (%)	40.0	35.0–45.0	42.5	40.0–50.0	0.277
SBP (mmHg)	130.0	112.0–159.0	133.0	119.0–158.0	0.249
DBP (mmHg)	85.5	71.0–98.3	82.5	70.0–92.0	0.542
Heart rate (bpm)	79.0	72.8–90.0	76.0	63.5–89.5	0.396
QTc (ms)	471.0	442.5–519.3	455.0	432.0–480.0	0.215
Creatinine (mg/dL)	0.8	0.7–1.0	0.7	0.7–0.9	0.010
LDL (mg/d)	100.0	70.3–137.0	90.5	76.5–120.5	0.695
CRP (mg/L)	0.6	0.4–3.9	0.5	0.1–0.8	0.267
HbA1c (%)	5.9	5.5–5.9	5.4	5.1–5.5	0.003
(hs)Troponin (pg/mL)	178.0	51.0–837.0	198.0	61.0–422.5	0.828
Pro-BNP (pg/mL)	3789.5	729.9–5816.8	2709.0	459.7–4999.5	0.361

AF (Atrial Fibrillation), BMI (Body Mass Index), CRP (C Reactive Protein), DBP (Diastolic Blood Pressure), EF (Ejection fraction), HbA1c (hemoglobin A1c), SBP (Systolic Blood Pressure), LDL (low density lipoprotein).

**Table 3 biomedicines-10-00464-t003:** Comparison of parameters of clinical presentation, precipitating factors, cardiovascular risk factors and demographics between patients with and without adverse cardiac events by Mann–Whitney-U test or if nominal by Chi Square test.

	Adverse Cardiac Events (*n* = 25)		Without Adverse Cardiac Events (*n* = 31)		*p* =
	Median or n (%)	IQR	Median or n (%)	IQR	
Betablockers	52.0		19.4		0.010
ACE/AT-1 inhibitor	56.0		32.3		0.074
Anticoagulation	20.0		9.7		0.272
SSRI/SNRI	28.0		19.4		0.446
Diuretics	24.0		12.9		0.217
Inhalation therapy	28.0		19.4		0.346
Age (years)	76.0	64.5–81.0	67.0	60.0–76.0	0.045
BMI (kg/m^2^)	25.8	21.8–28.9	25.0	21.6–30.2	0.972
SBP (mmHg	132.0	119.3–148.5	130.0	116.0–162.8	0.760
DBP (mmHg)	85.0	70.0–93.0	80.0	69.0–96.0	0.625
Creatinine (mg/dL)	0.8	0.7–1.0	0.7	0.7–0.9	0.010
LDL (mg/dL)	83.5	72.8–115.0	104.5	77.0–135.3	0.236
CRP (mg/L)	0.6	0.4–3.9	0.5	0.1–0.8	0.267
HbA1c (%)	5.5	5.2–5.9	5.4	5.3–5.7	0.932
Smoking	32.0		25.8		0.610
Hypertension	72.0		74.2		0.110
History of stroke	4.0		9.7		0.205
History of AF	8.0		9.7		0.251
EF (%)	40.0%	35.0–44.0%	45.0%	40.0–50.0%	0.019
Atrial size (ml)	42.0	31.0–70.0	40.0	34.0–69.3	0.981
E/E’	10.0	8.0–13.0	12.5	8.5–14.0	0.525
QTc (ms)	458.0	439.0–476.0	455.5	434.3–498.3	0.554
Heart rate (bpm)	82.0	73.0–90.0	75.0	61.0–83.0	0.112
Pro-BNP (pg/mL)	4299.5	1291.3–7684.3	2161.0	362.0–4222.0	0.026
hs-troponin (pg/mL)	225.0	69.0–843.0	186.0	53.0–369.0	0.179
Creatinine (mg/dL)	0.8	0.7–0.9	0.8	0.7–0.9	0.888
CRP levels (mg/L)	0.6	0.6–1.3	0.5	0.1–0.8	0.402

AF (Atrial Fibrillation), BMI (Body Mass Index), CRP (C Reactive Protein), DBP (Diastolic Blood Pressure), E/E’ (ratio between early mitral inflow velocity and mitral annular early diastolic velocity), EF (Ejection fraction), HbA1c (hemoglobin A1c), SBP (Systolic Blood Pressure), LDL (low density lipoprotein), SSRI/SNRI (Selective Serotonin Reuptake Inhibitor/selective noradrenaline reuptake inhibitor).

**Table 4 biomedicines-10-00464-t004:** Univariate and multivariate binary logistic of parameters of clinical presentation, precipitating factors, cardiovascular risk factors, and pretreatment with the risk of all-cause complications.

	All-cause Complications			All-Cause Complications		
	Univariate HR			Multivariate HR		
	B	95% CI	*p* =	B	95% CI	*p* =
Age (years)	1.053	0.997–1.113	0.062	1.034	0.973–1.100	0.281
BMI (kg/m^2^)	1.012	0.911–1.125	0.826			
Creatinine (mg/dL)	1.143	0.137–9.546	0.902			
CRP (mg/L)	0.985	0.831–1.167	0.859			
(hs)-troponin (pg/mL)	1.002	1.000–1.004	0.028			
Pro-BNP (pg/mL)	1.002	1.000–1.000	0.048			
EF (%)	0.922	0.854–0.995	0.036	0.925	0.854–1.003	0.058
Betablockers (n)	4.514	1.377–14.797	0.013	4.020	1.075–15.031	0.039
ACE/AT-1 inhibitor (n)	2.673	0.898–7.959	0.077			
Anticoagulation (n)	0.429	0.092–2.003	0.282			
Diuretics (n)	0.420	0.103–1.708	0.225			
SSRI/SNRI (n)	1.620	0.465–5.641	0.448			
Inhalation therapy (n)	0.549	0.156–1.930	0.350			

BMI (Body Mass Index), CRP (C Reactive Protein), DBP (Diastolic Blood Pressure), EF (Ejection fraction), HbA1c (hemoglobin A1c), SBP (Systolic Blood Pressure), LDL (low density lipoprotein), SSRI/SNRI (Selective Serotonin Reuptake Inhibitor/selective noradrenaline reuptake inhibitor).

**Table 5 biomedicines-10-00464-t005:** Comparison of parameters of clinical presentation, precipitating factors, cardiovascular risk factors, and demographics between patients with and without cardiac decompensation by Mann–Whitney-U test or if nominal by Chi Square test.

	Cardiac Decompensation (*n* = 20)		Without Cardiac Decompensation (*n* = 36)		*p* =
	Median or n (%)	IQR	Median or n (%)	IQR	
Betablockers	52.0		19.4		0.013
ACE/AT-1 inibitor	56.0		32.3		0.053
Anticoagulation	20.0		9.7		0.362
SSRI/SNRI	28.0		19.4		0.119
Diuretics	24.0		12.9		0.069
Inhalation therapy	30.0		19.4		0.342
Age (years)	76.0	64.5–81.0	67.0	60.0–76.0	0.252
BMI (kg/m^2^)	25.8	21.8–28.9	25.0	21.6–30.2	0.184
SBP (mmHg)	132.0	119.3–148.5	130.0	116.0–162.8	0.661
DBP (mmHg)	85.0	70.0–93.0	80.0	69.0–96.0	0.697
HbA1c (%)	5.5	5.2–5.9	5.4	5.3–5.7	0.526
Smoking	32.0		25.8		0.860
History of stroke	12.0		3.2		0.333
Oncologic disease	12.0		12.9		0.206
Female sex	92.0		93.5		0.699
EF (%)	40.0	35.0–45.0	45.0	40.0–50.0	0.074
LA (ml)	42.0	28.0–75.0	40.0	34.0–65.5	0.921
E/E‘	10.0	8.0–12.0	12.5	8.5–14.5	0.138
QTc (ms)	458.0	439.0–476.0	455.5	434.3–498.3	0.426
Heart rate (bpm)	82.0	73.0–90.0	75.0	61.0–83.0	0.300

AF (Atrial Fibrillation), BMI (Body Mass Index), CRP (C Reactive Protein), DBP (Diastolic Blood Pressure), E/E’ (ratio between early mitral inflow velocity and mitral annular early diastolic velocity), EF (Ejection fraction), HbA1c (hemoglobin A1c), SBP (Systolic Blood Pressure), LA (Left atrium), LDL (low density lipoprotein), SSRI/SNRI (Selective Serotonin Reuptake Inhibitor/selective noradrenaline reuptake inhibitor).

**Table 6 biomedicines-10-00464-t006:** Comparison of parameters of clinical presentation, precipitating factors, cardiovascular risk factors and demographics between patients with and without arrythmia by Mann–Whitney-U test or if nominal by Chi Square test.

	Arrythmia (*n* = 10)		Without Arrythmia (*n* = 46)		*p* =
	Median or n (%)	IQR	Median or n (%)	IQR	
Betablockers	26.3		13.5		0.236
ACE/AT-1 inhibitor	30.0		45.7		0.365
Anticoagulation	30.0		10.9		0.117
SSRI/SNRI	20.0		23.9		0.791
Diuretics	0.0		21.7		0.112
Inhalation therapy	10.0		26.1		0.319
Age (years)	61.5	56.0–78.8	72.0	63.0–78.0	0.252
BMI (kg/m^2^)	22.7	20.5–26.2	26.7	21.9–29.8	0.184
SBP (mmHg)	132.0	122.3–165.5	130.0	113.8–156.0	0.661
DBP (mmHg)	83.0	75.0–93.0	82.0	70.0–92.5	0.697
Creatinine (mg/dL)	0.7	0.6–0.7	0.8	0.7–0.9	0.077
LDL (mg/dL)	82.5	61.0–116.8	98.5	76.5–131.5	0.285
CRP (mg/L)	0.4	0.1–0.7	0.5	0.2–1.3	0.215
HbA1c (%)	5.4	5.1–5.8	5.5	5.3–5.8	0.526
Smoking	5.4		28.3		0.912
Hypertension	80.0		71.7		0.593
EF (%)	40.0	32.5–43.0	41.5	36.3–49.0	0.199
Atrial size (ml)	46.0	38.0–75.0	40.0	29.3–52.8	0.319
E/E’	11.4	8.8–19.0	11.0	8.0–13.3	0.384
QTc (ms)	456.0	408.8–484.0	457.0	438.5–490.5	0.426
Heart rate (bpm)	73.0	64.5–81.0	79.5	67.8–90.0	0.300
Pro-BNP (pg/mL)	2652.0	741.5–7266.5	3308.0	499.1–4919.8	0.711
hs-troponin (pg/mL)	145.0	42.5–485.3	205.5	70.5–465.0	0.521

AF (Atrial Fibrillation), BMI (Body Mass Index), CRP (C Reactive Protein), DBP (Diastolic Blood Pressure), E/E’ (ratio between early mitral inflow velocity and mitral annular early diastolic velocity), EF (Ejection fraction), HbA1c (hemoglobin A1c), SBP (Systolic Blood Pressure), LA (Left atrium), LDL (low density lipoprotein), SSRI/SNRI (Selective Serotonin Reuptake Inhibitor/selective nor-adrenaline reuptake inhibitor).

## Data Availability

Available from corresponding author upon reasonable request.

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
