# Peer review of "Pretreatment with Betablockers, a Potential Predictor of Adverse Cardiovascular Events in Takotsubo Syndrome"

_biomedicines, 2022, doi:10.3390/biomedicines10020464_

Round 1
Reviewer 1 Report
The subject of the work is very interesting and clinically important, but the manuscript requires many corrections.
The abstract needs improvement, mainly methods (grouping, observation period) and results (more concrete data with numerical results).
The small study group is a limitation, as the authors noted in Limitations.
Methods
What was the enrollment period for the study? Were the Mayo Clinic Criteria in force at the time? What was the observation period? Please describe the division into study groups.
Results
Previous use of beta-blockers was likely to induce other cardiovascular diseases in the patients. Please complete Table 2 with comorbidities, e.g. coronary artery disease, hypertension, previous MI, stroke, arrhythmias, etc.
The presentation of the results requires ordering in the text and tables. I propose to present the results in groups: demographic data (age, sex, BMI), comorbidities, clinical parameters, laboratory data, etc. Please transfer the "sex" line between "age" and "BMI" in tables 1 and 2.
Tables - no explanation of the abbreviations used under the tables Table 2 - The title of the table 2 is on a separate page.
The text does not require precise figures, which can be found in the tables.
For data presented only in the text, add percentages.
In the manuscript, the authors often use the term "forced diuresis"? I believe that the term "diuretic drug use" is better.
Author Response
Reviewer 1
Comment 1:
The subject of the work is very interesting and clinically important, but the manuscript requires many corrections.
The abstract needs improvement, mainly methods (grouping, observation period) and results (more concrete data with numerical results).
The small study group is a limitation, as the authors noted in Limitations.
Response 1: We thank the reviewer for the productive evaluation of our manuscript. The abstract and the methods were fundamentally corrected to better explain the structure and the endpoints of the study.
Comment 2: What was the enrollment period for the study? Were the Mayo Clinic Criteria in force at the time? What was the observation period? Please describe the division into study groups.
Response 2: Information about the period of the enrollment, about the observation period and the division into study groups was added in the abstract and the methods of the revised manuscript. (line 67-76) The Mayo Clinic Criteria were always in force at the time.
Comment 3:
Previous use of beta-blockers was likely to induce other cardiovascular diseases in the patients. Please complete Table 2 with comorbidities, e.g. coronary artery disease, hypertension, previous MI, stroke, arrhythmias, etc.
The presentation of the results requires ordering in the text and tables. I propose to present the results in groups: demographic data (age, sex, BMI), comorbidities, clinical parameters, laboratory data, etc. Please transfer the "sex" line between "age" and "BMI" in tables 1 and 2.
Tables - no explanation of the abbreviations used under the tables Table 2 - The title of the table 2 is on a separate page.
The text does not require precise figures, which can be found in the tables.
For data presented only in the text, add percentages.
Response 3: Table 2 was upgraded with cardiovascular comorbidities including coronary artery disease, hypertension, stroke and arrythmias. There is however an increase in comorbidities, hypertension and atrial fibrillation, in past medical history in patients taking a preadmission betablocker. The difference is related to therapeutic applications for beta blockers and was not significant based on endpoints of our study.
The title of table was placed on the same page and abbreviations were added.
The structure of the tables and the text was changed according to the reviewer’s recommendation. Information, already found in the tables, was deleted in the text.
Comment 4: In the manuscript, the authors often use the term "forced diuresis"? I believe that the term "diuretic drug use" is better.
Response 4: The term “forced diuresis” was replaced by diuretic drug use.
Reviewer 2 Report
In the paper "Pretreatment with beta-blockers, a potential predictor of adverse cardiovascular events in Takotsubo syndrome" the authors analyzed the effects of preadmission beta-blocker use on adverse events in patients with TTS
The topic could be of some interest, however, some points need to be addressed
1) English requires improvement prior to publication, please carefully revised the manuscript in order to improve readibility
2) It's not clear the reason for beta-blockers therapy in patients enrolled in the study: This point is very important as the disease for which beta-blockers is described could justify the results of the study
3) How is defined congestion? Please specify
4) what kind of arrhythmias are considered secondary end-points? All supraventricular and ventricular arrhythmias, only some type?
5)the therapy with beta-blockers was continued after TTS diagnosis?
Author Response
Reviewer 2
Comment 1: In the paper "Pretreatment with beta-blockers, a potential predictor of adverse cardiovascular events in Takotsubo syndrome" the authors analyzed the effects of preadmission beta-blocker use on adverse events in patients with TTS
The topic could be of some interest; however, some points need to be addressed
Response 1: We thank the reviewer for his/her kind remarks and enthusiasm for our work.
Comment 2: English requires improvement prior to publication, please carefully revised the manuscript in order to improve readability
Response 2:
The manuscript was linguistically revised by a native English speaker.
Comment 3: It's not clear the reason for beta-blockers therapy in patients enrolled in the study: This point is very important as the disease for which beta-blockers is described could justify the results of the study
Response 3: Betablockers were most frequently used as an adjunct hypertensive drug therapy or as frequency control therapy in patients with atrial fibrillation. However, hypertension or atrial fibrillation, as a patient’s characteristics did not affect the risk of adverse cardiac events. (line 28-33; line 131-135; line 276-280)
Comment 4: How is defined congestion? Please specify
Response 4: The endpoint of cardiac decompensation was specified in the revised manuscript. (line 90-91)
Comment 5: What kind of arrhythmias are considered secondary end-points? All supraventricular and ventricular arrhythmias, only some type?
Response 5: We thank the reviewer for this important input. The inclusion criteria for the endpoint of arrythmia were specified in the methods. More precise information about the occurrence of arrythmias was reported in the revised part of the results. (line 82-87)
Comment 6: the therapy with beta-blockers was continued after TTS diagnosis?
Response 6: Patients on pretreatment with betablockers remained on therapy until discharge. This important information was added in the revised manuscript. (line 75)
Reviewer 3 Report
In this observational study, Dr. Topf and colleagues studied the adverse effects of beta-blocker pretreatment in Takotsubo / stress-induced cardiomyopathy. Overall, it is a nice study with some clinical insights. However, some unclarities were found and need to be addressed by the authors:
- Since the major concern with beta-blocker pretreatment in TTS patients is the worsening of the clinical condition, leading to death, I think it is important to add a section about its effect on (all-cause / CV-related) mortality. Please add it as a new section and perform all necessary analyses to identify the risk comparing patients with and without beta-blocker.
- "The effect of preadmission betablocker use on adverse events has not been studied in the literature." adverse events of what?
- "To investigate this issue, we evaluated clinical complications in 56 patients hospitalized for TTS" clinical complications of what?
- It remains unclear how the authors selected the individuals opted in this study, especially the ones with beta-blocker pretreatment. Please add this information in the abstract and methods.
- Why didn't the authors select equal number of individuals with and without beta-blocker pretreatment? Are the authors sure that there was no significant impact of sample size imbalance?
- "There is some evidence for beneficial effects..." I think this is grammatically incorrect.
- "Patients were enrolled if they fulfilled the Mayo Clinic Diagnostic Criteria for TTS" Please elaborate the criteria in this manuscript. Use a table if needed.
- There was no explanation about the beta-blocker therapy at all in the methods. Please provide the details of the pre-treatment. How long? since when? continuous or intermittent? which beta-blocker (with ISA or not, selective or not, mention the name)? the dose? There are many things that could affect the interpretability of the findings so please complete these data. Use a table if needed.
- "As expected, TTS patients showed a reduced left ventricular ejection fraction on admission" Who expected the LVEF reduction? TTS could also be asymptomatic so please remove "as expected"
- I think it is better to also use tables to better display the findings in section 3.3 and 3.4. Consider adding.
- What does it mean by "all cause complications" and "adverse cardiac events"? What are the components? Please define somewhere in the methods or result section.
- Please also add the univariate and multivariate binary logistic of parameters for cardiac decompensation (section 3.4).
- Similarly, add all the complete data of section 3.5 about arrhythmia similar to what have been done in section 3.3 and 3.4, although it was not significantly different. Also, add the above-requested table and univariate/multivariate analyses for section 3.5.
- Line 178: "arrhythmia" fix the typo
- "Based on our results, we conclude that preadmission betablocker therapy increased the risk of adverse cardiac events 4.5-fold in our study population." Is there any specific reason for using the univariate result, not the multivariate one?
- Also, please add the fold of increased risk of decompensation by beta-blocker pretreatment in the conclusion.
- "Among the limitations of the present study are the small cohort. Large-scale studies are required to confirm the results of the present study." Please make sure to declare ALL limitations of this study and provide some recommendations for future studies on the same topic.
Author Response
Reviewer 3
Comment 1: In this observational study, Dr. Topf and colleagues studied the adverse effects of beta-blocker pretreatment in Takotsubo / stress-induced cardiomyopathy. Overall, it is a nice study with some clinical insights. However, some unclarities were found and need to be addressed by the authors:
Response 1: We thank the reviewer for his/her kind remarks and enthusiasm for our work.
Comment 2: Since the major concern with beta-blocker pretreatment in TTS patients is the worsening of the clinical condition, leading to death, I think it is important to add a section about its effect on (all-cause / CV-related) mortality. Please add it as a new section and perform all necessary analyses to identify the risk comparing patients with and without beta-blocker.
Response 2: There were no fatalities among the patients included in this study. (line 129)
Comment 3: "The effect of preadmission betablocker use on adverse events has not been studied in the literature." adverse events of what?
Response 3: The sentence was corrected. (line 16)
Comment 4: "To investigate this issue, we evaluated clinical complications in 56 patients hospitalized for TTS" clinical complications of what?
Response 4: The sentence was specified in the revised manuscript. (line 18)
Comment 5: It remains unclear how the authors selected the individuals opted in this study, especially the ones with beta-blocker pretreatment. Please add this information in the abstract and methods.
Response 5: The inclusion of the patients was specified in the revised methods and in the abstract. (line 71-75)
Comment 6: Why didn't the authors select equal number of individuals with and without beta-blocker pretreatment? Are the authors sure that there was no significant impact of sample size imbalance?
Response 6: We thank the reviewer for this important comment. The unequal distribution between the two groups is a bias. This information was added in the limitation (line 333-337).
Comment 7: "There is some evidence for beneficial effects..." I think this is grammatically incorrect.
Response 7: The sentence was corrected in the revised manuscript.
Comment 8: "Patients were enrolled if they fulfilled the Mayo Clinic Diagnostic Criteria for TTS" Please elaborate the criteria in this manuscript. Use a table if needed.
Response 8: The Mayo Clinic Diagnostic Criteria were added in the methods part. (line 69)
Comment 9: There was no explanation about the beta-blocker therapy at all in the methods. Please provide the details of the pre-treatment. How long? since when? continuous or intermittent? which beta-blocker (with ISA or not, selective or not, mention the name)? the dose? There are many things that could affect the interpretability of the findings so please complete these data. Use a table if needed.
Response 9: “Pretreatment with betablockers” was defined in the abstract and the methods. (line 73). Additionally, we gave information about the type of betablocker and the dosage. (line 131-135)
Comment 10: "As expected, TTS patients showed a reduced left ventricular ejection fraction on admission" Who expected the LVEF reduction? TTS could also be asymptomatic so please remove "as expected"
Response 10: The sentence was corrected in the revised manuscript.
Comment 11: I think it is better to also use tables to better display the findings in section 3.3 and 3.4. Consider adding.
Response 11: We thank the reviewer for this important input. Tables were added to better visualize the results.
Comment 12: What does it mean by "all cause complications" and "adverse cardiac events"? What are the components? Please define somewhere in the methods or result section.
Response 12: All cause complications and adverse cardiac events were used synonymously throughout the manuscript. This important information was added in the methods. (line 94)
Comment 13: Please also add the univariate and multivariate binary logistic of parameters for cardiac decompensation (section 3.4).
Response 13: A univariate and multivariate binary logistic analysis was added. Whereas univariate/multivariate analysis for the major endpoint of adverse cardiac events of our study remained significant in our study, post hoc multivariate analysis for the association of prior betablocker therapy with the risk of cardiac decompensation remained slightly unmet in a multivariate regression model after correction for age and systolic left-ventricular ejection fraction (p=0.073).
Comment 15: Similarly, add all the complete data of section 3.5 about arrhythmia similar to what have been done in section 3.3 and 3.4, although it was not significantly different. Also, add the above-requested table and univariate/multivariate analyses for section 3.5.
Response 15: A table, depicting differences between TTC patients with and without arrythmia was added as a suppl. Table in the revised manuscript. In univariate and multivariate analysis, there was no statistically significant difference.
Comment 15: “Based on our results, we conclude that preadmission betablocker therapy increased the risk of adverse cardiac events 4.5-fold in our study population." Is there any specific reason for using the univariate result, not the multivariate one?
Also, please add the fold of increased risk of decompensation by beta-blocker pretreatment in the conclusion.
Response 15: Missing information was added in the revised manuscript. (line 328)
Comment 16: "Among the limitations of the present study are the small cohort. Large-scale studies are required to confirm the results of the present study." Please make sure to declare ALL limitations of this study and provide some recommendations for future studies on the same topic.
Response 16: The unequal distribution between the two groups was added as another limitation. Further limitation is the unequal distribution of the patients’ characteristics hypertension. (line 333-337)
Round 2
Reviewer 2 Report
No other concern.
Author Response
We thank the reviewer for the productive evaluation of our manuscript.
Reviewer 3 Report
Thanks for addressing my previous comments adequately. I have no further remarks.
Author Response
We thank the reviewer for the productive evaluation on our manuscript.